# Assessing Impacts of Additives on Particulate Matter and Volatile Organic Compounds Produced from the Grilling of Meat

**DOI:** 10.3390/foods11060833

**Published:** 2022-03-14

**Authors:** Xingyun Liu, Wei Xing, Zhaoyang Xu, Xiaomin Zhang, Hui Zhou, Kezhou Cai, Baocai Xu, Conggui Chen

**Affiliations:** 1Engineering Research Center of Bio-Process, Ministry of Education, Hefei University of Technology, Hefei 230009, China; luckin961229@163.com (X.L.); xw18269791273@163.com (W.X.); 911098854qq@gmail.com (Z.X.); 17855301709@163.com (X.Z.); zhouhui@hfut.edu.cn (H.Z.); baocaixu@163.com (B.X.); chencg1629@hfut.edu.cn (C.C.); 2Key Laboratory for Agricultural Products Processing of Anhui Province, School of Food Science and Bioengineering, Hefei University of Technology, Hefei 230009, China

**Keywords:** volatile organic compounds, particulate matter, carbonyl compounds, grilling of meat, additives

## Abstract

Cooking fumes are an important source of volatile organic compounds (VOCs), particulate matter (PM), and carbonyl compounds. The additive is wildly applied in grilling meat for flavor improvement. However, the effects of additives on cooking fumes emissions, such as volatile organic compounds (VOCs), particulate matter (PM), and carbonyl compounds, in meat grilling have not been studied. The impact of four additives, including white pepper, salt, garlic powder, and compound marinade, on the emission characteristics of cooking fumes from the grilling meat was investigated. The concentrations of VOCs and carbonyl compounds in the cooking fumes were analyzed by TD-GC/MS and HPLC, respectively. The PM emission characteristics (mass concentration and size distribution) were measured by DustTrak DRX aerosol monitor in real-time. Results showed that the application of white pepper, salt, garlic powder, and mixed spices could significantly reduce the total particles mass concentration (TPM) emissions during meat-grilling by 65.07%, 47.86%, 32.87%, and 56.01%, respectively. The mass concentration of PM during meat-grilling reached maximum values ranging from 350 to 390 s and gradually fell at the final stages of grilling. The total concentration of 22 representative VOCs emitted from the grilling was significantly increased in grilling meat marinated with compound additives. Aromatic hydrocarbons were the predominant VOCs species, followed by ketone compounds. During the grilling process, formaldehyde, acetaldehyde, propionaldehyde, and acetone were major carbonyl compounds. The low molecular weight carbonyl compounds (C1–C3) in cooking fumes were dominant carbonyl compounds.

## 1. Introduction

The cooking fumes emitted from cooking activities result in human exposure to particulate matter (PM) and volatile organic compounds (VOCs) [1]. The VOCs species are a complex mixture of hazardous chemicals, which not only consists of benzene series, alkanes, chlorinated VOCs, oxygenated VOCs, and alkenes, but also carbonyl compounds [2]. Previous studies showed that most carcinogenic polycyclic aromatic hydrocarbons (PAHs) in cooking fumes are absorbed onto PM, especially particles with aerodynamic equivalent diameters less than 0.43 µm [3]. Several studies have also proved that PM and VOCs emitted from cooking activities consist of multiple hazardous chemical compounds which are easily deposited in the alveoli and can cause cardiovascular and respiratory disease or even death. Apart from these negative effects, cooking ultrafine particles can also impact human brain activity [4,5,6,7,8,9]. Therefore, characterizing VOCs, PM, and its chemical constituent and exploring beneficial methods for reducing the emission of cooking fumes are significant in improving the healthy environment of food processing or cooking for human beings.

The grilling process, which suffers a higher temperature, has been found to generate dominant cooking fume emissions compared with other cooking methods [10]. During thermal cooking activities, a lot of complex chemical reactions are involved in the grilling process. The carbohydrates or sugars (including oligosaccharides, disaccharides) that exist in the food undergo hydrolysis with water during the process of thermal cooking. When the mixture is continuously heated, degradation chemical reactions will take place, and the rings of sugar will open up to generate new molecules, such as aldehyde compounds and acids [11]. With the adequate increase in temperature, the products of degradation reactions will recombine to generate chain-like molecules. Moreover, the products of degradation reactions (acids and aldehyde compounds) could react with amino acids to form a lot of VOCs [12].

The early publications showed that the emission characteristics of PM and VOCs during cooking activities could be influenced by many factors. Torkmahalleh et al. [13] studied the effect of additives on the emission characterization of PM_2.5_ and total particle number during the heating of cooking oils. The results indicated that the addition of sea salt could reduce the PM_2.5_ concentration by 86–91% and total particle number by 45–53% compared with the control group. Katragadda et al. [14] assessed the impacts of oil types on volatile aldehydes emissions produced from heated cooking oils. They found that the emission of volatile compounds was significantly higher in extra virgin olive oil. Zhang et al. found that ventilation systems could impact the decay rate of PM emissions from various cooking methods [15]. These results show that properly controlling methods, such as adding suitable additives, selecting the low emitting oil type, or application of ventilation during cooking activities, could reduce the PM mass concentration and VOCs emissions.

Additives are widely applied in different cooking methods. The synergistic effect of spice additives and different cooking methods can cause the formation of a characteristic aroma via induced reactions that ameliorate the aroma profiles of products. The key aroma compounds will be generated in the thermal process of cooking activities due to a wide range of complex chemical reactions involving lipid oxidation and pyrolysis reactions, thiamine degradation, proteolysis reactions, Maillard reaction, and Maillard-lipid interactions [16,17,18,19]. The addition of various additives can improve the formation of key aroma compounds to change the quality of meat products and consumer acceptability. Black pepper, turmeric, salt, and garlic powder are commonly used additives in many cooking foods, especially in grilling food. However, the impacts of these additives on the PM and VOCs produced by grilling are still unknown. The objective of the present study is to systematically assess the effect of marinating with these additives on PM and VOCs emissions in cooking oil fumes during grilling meat. This study can provide guidance on choosing the proper combination of additives for the best cooking activities, in terms of minimizing PM and VOCs emissions and effectively alleviating the pressure of ultimate purification.

## 2. Materials and Methods

### 2.1. Materials and Sampling Instruments

Four additives, including salt, white pepper, and garlic powder, are commonly used in the grilling of meat and were purchased from a local supermarket in Hefei, China. Pork belly and high-density bamboo charcoal (HDBC) were also purchased from a local Carrefour supermarket (Hefei, China). In addition, by mixing the salt, white pepper, and garlic powder by a ratio of 1:2:2, the mixed spices (MS) were made to marinate the meat.

Particle mass concentration and size distribution were monitored with a TSI Model 8533 DustTrak-DRX Aerosol Monitor (St. Paul, MN, USA), which has PM_1.0_, PM_2.5_, PM_4.0_, PM_10_, and TPM (total particles mass concentration) inlets. Although the DustTrak Aerosol Monitor captures only a limited portion of PM mass concentration, the aerodynamic equivalent diameter of particulate matter greater than 500 nm constitutes the majority of the estimated PM mass concentration [20]. A JCH-2400 dual-channel constant current air sampler (Qingdao Juchang Environmental Protection Group Co., Ltd., Qingdao, China) was also used as a sampling pump to sample the carbonyl compounds.

### 2.2. PM and VOCs Emission Experiments for Different Additives

Four additives were selected for the marinade: white pepper powder, garlic powder, salt, and mixed spices. The pork was pre-marinated with additives at a ratio of 0.5 g/100 g, respectively. Then, the marinated meat was transferred to a low-temperature cold store and marinated for 6 h. As a control, the same pork was used without any spice additive. The marinated pork was grilled using a round grilling plate. The HDBC was used as the fuel of the meat-grilling process and transferred to the charcoal basin after fully burning for 5 min. A thermocouple was placed above the grilling plate to monitor the heating temperature, and the variations of temperature were less than 6 °C. When the temperature of the grilling plate reached 320 °C, the simulation meat-grilling started.

In order to determine the emission characteristics of PM and VOCs objectively and scientifically during the meat-grilling in the presence of different additives, our group have established the simulation and detection platform for cooking fumes emitted from meat-grilling. The meat-grilling experiments were conducted at Hefei University of Technology, Feicui Lake campus. The cooking fumes emitted from the grilling were captured by a 40-inch by 50-inch stainless steel hood and ducted to the stack of the facility with an exhaust fan. The exhaust fan had a variable speed drive and controller, which was used to adjust the velocity and flow rates through the stack. The experiments system used to conduct this study is shown in Figure 1.

Prior to the experiments, no cooking activities other than meat-grilling took place. Background samples were monitored before the meat-grilling event in the simulated kitchen and were determined for correction of the presence of particulate matter, aldehyde and ketone compounds, and volatile organic compounds that were not related to the simulation experiment. In addition, the high-density bamboo charcoal (HDBC) was selected as the fuel for meat-grilling. The HDBC we used in the simulation experiment was from the same batch with the same quality, which was mixed evenly.

### 2.3. Determination of the Chemical Composition of the Pork

As for moisture content, lipid content, and protein content of raw pork and marinated meat, each batch was determined by the following method. Firstly, moisture content: the determination of the moisture content in pork mainly referred to the direct drying method in the GB/T 5009.3-2016 [21]. The experimental procedures are as follows: firstly, leave the cover off the weighing bottle and put them into a drying oven until dry to a constant weight. The constant weight of the weighing bottle was recorded to M_0_. Then, the 5.00 g minced meat of pork belly was weighed and put into the weighing bottle; the total weight of meat and the weighing bottle was recorded to M_1_. Secondly, place the sample and weighing bottle in a 105 °C drying oven until dry to a constant weight. After being taken out, they should be placed into a dryer to cool for 30 min. The constant weight of the sample and weighing bottle was recorded to M_2_. The moisture content should be calculated according to the formula:Moisture content (%) = M_2_ − M_0_/M_1_ − M_0_ × 100%

For the lipid content: the determination of lipid content in pork mainly referred to GB/T 5009.6-2016 [22]: the soxhlet extractor method.

For the protein content: the determination of the protein content in pork mainly referred to GB/T 5009.5-2016, i.e. the Kjeldahl nitrogen determination method [23]. The determination of the protein content was adjusted in combination with the instrument instructions of the Kjeldahl nitrogen determination instrument.

The experimental data moisture content, lipid content, and protein content are shown in Appendix A.

### 2.4. Particulate Matter Analysis

The PM mass concentration and size distribution emitted from the grilled meat were analyzed using the light-scattering method [24]. The whole processing time of grilling was 420 s. The PM mass concentration, the size distribution of PM less than 1.0 μm (PM_1.0_), PM less than 2.5 μm (PM_2.5_), PM less than 4.0 μm (PM_4.0_), PM less than 10.0 μm (PM_10_), and total concentration were measured using a DustTrak Aerosol Monitor (8533, TSI, St.Paul, MN, USA) equipped with an electrostatic prevention hose. The electrostatic prevention hose was installed at the sampling site to monitor the PM concentration (Figure 1). The electrostatic prevention hose and Aerosol Monitor were connected, and the sampling intervals were set to 5 s which corresponded to 1 scan record. The sampling air flow rate was 500 mL/min, and each simulation experiment was tested for 15 min.

### 2.5. Aldehyde and Ketone Compounds Analysis

During meat grilling, the produced cooking oil fumes passed through the purification system, and the carbonyl compounds were sampled on a silica cartridge impregnated with 2,4-DNPH (Sep-Pak DNPH-Silica Cartridges Plus-Short Body, Waters, Milford, MA, USA). The cooking fumes during the meat-grilling was drawn into the 2,4-dinitrophenylhydrazine-silica cartridge for the derivatization of the DNPH–Aldehyde compounds using a dual-channel constant current air sampler with a sampling flow rate of 500 mL/min. Each sample was collected for 30 min. A PTFE filter was set in the front of the sampling cartridge to remove the PM cooking fumes and a short ozone scrubber (Sep-Park, Waters, Milford, MA, USA) was connected to the inlet of the DNPH–Silica cartridges to prevent interference from ozone in the cooking fumes. After complete sampling, all sampled cartridges were stored in 4 °C refrigerators in sealed aluminum bags until extraction. Each sampler was eluted with 10 mL acetonitrile (HPLC Grade) solution and transferred to a 10-mL volumetric flask. The samples were analyzed to determine aldehyde and ketone concentrations by high-performance liquid chromatography (Model 1260, Agilent Inc., Santa Clara, CA, USA) with a Symmetry ^®^ C18 4.6 × 250 mm column (Waters, Wexford, Ireland) and a UV detector at 360 nm. The injection volume was 20 μL, and the gradient mobile phase consisted of acetonitrile and water. The gradient program was performed at a flow rate of 1 mL/min, and detailed information on the gradient elution program is shown in Table 1. A total of 13 aldehydes and ketones were selected as target carbonyl compounds to be sampled and analyzed according to the HJ 683 (Ministry of Ecology and Environment of the People’s Republic of China, 2014) [25] (Table 2).

### 2.6. Volatile Organic Compounds (VOCs) Analysis

The VOCs emitted from meat-grilling were collected and analyzed according to HJ 734 (Ministry of Ecology and Environment of the China, 2014) [26] and Zhang et al. [27]. VOCs were collected using a stainless-steel tube (6 mm × 90 mm, PerkinElmer, Fremont, CA, USA) containing Carbopack C sorbent, Carbopack B sorbent, and Carboxen 1000 sorbent (60/80 mesh, Suplelco, St. Louis, MO, USA). Before sampling, the VOCs’ sorbent tube was cleaned at 320 °C for 30 min. The stainless-steel sorbent tube was activated using an adsorption tube activation apparatus (ACT-10, Ledon Technologies Ind., Suzhou, China). VOCs in the cooking fumes were collected in the stainless-steel sorbent tube using a dual-channel constant current air sampler (Qingdao Juchang Environmental Protection Group Co, Ltd., Qingdao, China). The sampling flow rate was 500 mL/min, and the total sampling time was 60 min. Then, the VOCs were thermally desorbed using a thermal desorption system (TD-100, Markers, Birmingham, UK) and determined using a Gas Chromatography (7890 A, Agilent, Santa Clara, CA, USA) coupled with a Mass Spectrometry (5975 C, Agilent, Santa Clara, CA, USA) with a capillary column (60.0 m × 0.25 mm × 1.40 µm, VF-624, Agilent). The initial column temperature was 35 °C for 5 min and then was raised to 140 °C at a rate of 6 °C/min. After that, the final column temperature was 220 °C with a rising rate of 15 °C/min and was maintained for 2 min. The mass spectrometer was performed by electron-impact ionization at an electron energy of 70 eV. The MS was based on the total ion scan mode, and the whole mass range was scanned at the frequency of 1.5 Hz. The thermal desorption apparatus constituted two desorption steps: (1) the temperature of primary desorption was 300 °C for 10 min, and the flux of desorption air was 30 mL/min; (2) the temperature of secondary desorption was also 300 °C for 4 min, and the flux of desorption air was same as (1). The desorbed VOCs were transferred to the GC-MS for determination [28].

The individual concentrations of the 22 representative VOCs were quantified using the standard compounds. The method was as follows: firstly, remove 25 μL, 50 μL, 100 μL, 250 μL, and 500 μL of the standard stock solution with a microsyringe to a 10-mL volumetric flask and dilute it with methanol to the marked line. The concentration gradients of 5, 10, 20, 50, and 100 μg/mL mixed standard solution were prepared. The adsorption tube on the thermal desorption standard sample loading platform was installed, and 1 μL of the mixed standard solution was injected with a microsyringe into the blank adsorption tube. Then, the internal standard solution was added to the adsorption tube at the same time, purging the adsorption tube with N_2_ for 5 min. The adsorption tube was removed and sealed at both ends with a sealing cap to obtain a calibration series of adsorption tubes with contents of 5, 10, 20, 50, and 100 ng. The adsorption tubes of the calibration curve series were put into the thermal desorption instrument and analyzed from low concentration to high concentration according to the test conditions. Then, the calibration curve was drawn with the least square method or relative response factor. The retention time and mass spectrum were compared for qualitative analysis and quantification of the 22 representative VOCs based on the standard curve.

### 2.7. Quality Control

The concentration of detected compounds in the DNPH–Silica cartridge blanks and stainless-steel sorbent tube were below the detection limits. The relative standard deviation of standard compounds was within 10%. In addition, the DustTrak Aerosol Monitor principle of working is light scattering which typically overestimates mass concentration, and the readings of the DustTrak monitors were calibrated against a gravimetric sampler. In order to correct the error of instrument monitoring, the gravimetric method is often used to measure the mass concentration of particulate matter. Due to the limitations of laboratory conditions, this was also a limitation of this research.

### 2.8. Statistical Analysis

All the experiments in this study were performed in triplicates. The data are expressed as mean ± standard deviation (SD). One-way analysis of variance (ANOVA) was conducted using SPSS 23.0 (SPSS Inc., Chicago, IL, USA), and statistical differences were performed using Tukey’s comparison test, with *p*-value < 0.05 regarded as significantly different. The experimental data were processed using Origin 2019 software (Origin Lab, Northampton, MA, USA).

## 3. Results and Discussion

### 3.1. Mass Concentration and Size Distribution of the PM during the Meat-Grilling Process

PM mass concentration emissions and size distribution for the group with additives and the control group are indicated in Table 3. The statistically significant differences were observed between marinated groups and the control. It was found that particulate matter ranged from different size distributions of emission concentrations, with the highest level in the control. The results indicated that all marinade-treatment with white pepper powder, salt, garlic powder, and MS can reduce the concentration of PM emissions compared to the control testing. The meat marinated with white pepper powder showed the largest reduction in PM mass concentration. Appendix A shows the average total particulate matter (TPM) mass concentration emitted from the meat-grilling process, and the TPM mass concentration was determined to be 40.47 ± 5.16 mg/m^3^ (control group), 21.1 ± 3.52 mg/m^3^ (salt), 14.13 ± 4.09 mg/m^3^ (white pepper), 27.17 ± 2.97 mg/m^3^ (garlic powder), and 17.8 ± 0.95 mg/m^3^ (MS). The meat without any additive marinade generated higher (*p* < 0.05) TPM mass concentration compared to the marinated meat with white pepper, salt, garlic powder, and MS during meat-grilling. Among these additives, white pepper, salt, garlic powder, and MS reduced the TPM emissions during meat-grilling by 65.07%, 47.86%, 32.87%, and 56.01%, respectively. These results indicate that marination with exogenous spice additives, especially with white pepper, is an effective method for decreasing the emission of TPM mass concentration during meat-grilling.

Table 4 shows the percentage of PM_2.5_/TPM emitted from the meat-grilling with different treatments. In this study, PM_2.5_/TPM of all groups ranging from 86.49 to 91.59% manifested that PM_2.5_ is the majority particulate matter of TPM. Previous reports determined the percentage of PM_2.5_ at the vent of cooking fumes in different restaurants, and the results were usually 0.6 [29]. Wan et al. [30] also showed that the size distribution of PM (<100 nm) generated from cooking contributed to about 75% of total particulate matter concentration. The results of PM_2.5_/TPM in this study are higher than those of previous studies, which may derive from the variation of the simulation experiment conditions.

Zhang et al. concluded that water-based cooking activities could produce less ultrafine particulate matter and PM_2.5_ than oil-based cooking methods [14]. In our present experiment, the skin of the chicken wing grilled was rich in fat, which would contribute to the forming of the ultrafine particulate matter and PM_2.5_. Additionally, four groups pre-marinated with additives containing more water led to a smaller ratio of PM_2.5_/TPM, which further supported the above conclusion.

### 3.2. Time Profiles of PM Emission during Meat-Grilling

Figure 2 presents time profiles of PM emissions during the meat-grilling process. As for the whole thermal process of grilling, it can be divided into three periods: initial stage, medium term, and final phase. The PM mass concentration showed a low emission level at the initial stage of grilling, and it reached maximum values from 350 to 390 s in the medium term. At this stage, the mass concentration of the particulate matter was much higher than the background concentration. During the medium stage of cooking activities, the average number of concentrations of UFPs in the kitchen was about 20–40 times the background level [30]. At the final phase, the mass concentration of PM began to decline. The variation trend of the particle mass concentration in the entire process of grilling shows that in the middle and late stage of cooking activities, humans will be exposed to a high concentration of PM which may result in considerable negative impacts on human health in indoor environments.

There was a time delay for PM emissions reaching maximum PM mass concentration during the meat-grilling process. It was a 150-s delay for the four kinds of particle dimensions, including PM_1.0_, PM_2.5_, PM_4.0_, and PM_10_. It can be observed that the PM concentration emitted from the meat-grilling reached a low-level concentration prior to 150 s except for control testing, then quickly increased to the maximum concentration of 68.3–99.43 mg/m^3^ at around 350–390 s. This observation was similar to the previous study with non-ventilation conditions [31]. 

### 3.3. Aldehyde and Ketone Compounds Emissions

The effect of additives on the aldehyde and ketone compounds (13 representative compounds) emissions is shown in Table 5. Statistically significant differences were observed in the total carbonyl compounds between groups marinated with salt, white pepper, and the control (*p* < 0.05). For treatment groups marinated with white pepper, salt, garlic powder, and mixed spices, the average total carbonyl compound concentrations were 7524.57 ± 218.16 μg/m^3^, 6694.37 ± 456.25 μg/m^3^, 7900.02 ± 74.43 μg/m^3^, and 8206.65 ± 269.55 μg/m^3^, respectively. The total aldehyde and ketone compounds were the highest emissions for the control group and reached 8331.33 ± 274.45 μg/m^3^ (Table 5).

As for the individual carbonyl compounds, formaldehyde, acetaldehyde, acetone, and propionaldehyde were the most abundant carbonyl compounds, with the percentages of total aldehyde and ketone compound concentrations at 24.40–34.01%, 15.09–20.13%, 16.85–27.86%, and 6.00–9.60%. Pork contains various unsaturated fatty acids and saturated fatty acids [32,33]. Linolenic acid was associated with the production of acetaldehyde, oleic acid was associated with the production of acetaldehyde and propanal, and palmitic acid was associated with the production of nonanal [12].

However, acrolein was not detected in all the cooking fumes emitted from meat-grilling. Benzaldehyde, valeraldehyde, butyraldehyde, and 2-butanone showed a low percentage of total carbonyl compounds, and the percentages ranged from 0.46% to 8.33%. Moreover, other carbonyl compounds were determined in the cooking fumes, and their percentages of total carbonyl compounds were mostly <10%. Aldehyde compounds are the dominant odorous compounds generated from cooking activities [34]. The aldehyde compounds are produced via the hydrolyzation of the hydrocarbons in food and via the oxidation of fatty acids [35]. Fullana et al. found that volatile aldehydes were generated from β-scission of alkoxy radicals formed by the cleavage of fatty acid hydroperoxides [36,37].

The relative proportions of C_1-8_ carbonyl compounds in the cooking fumes produced from the grilling are shown in Figure 3. The low molecular weight carbonyl compounds (C_1_–C_3_) in cooking fumes were dominant compounds in the grilling of meat. The process of meat-grilling mainly generated C_1_, C_2_, and C_3_ carbonyl compounds, accounting for 24.40–34.01%, 15.09–20.13%, and 25.54–37.46%, respectively. The higher concentration of C_1_–C_3_ compounds could be explained by the partial degradation of fatty acids to produce aldehyde and ketone compounds [38] and the incomplete combustion of meat [6]. The total aldehyde and ketone compound concentrations in our study were of a similar magnitude to the previous report by Ho et al. [34]. Ho proved that low molecular weight carbonyl compounds, such as acetone, acetaldehyde, and formaldehyde, were dominant compounds. In other research, C_1_–C_3_ carbonyl compounds were also the highest contributor, accounting for more than 80% of the total aldehyde and ketone compounds. Among the carbonyl compound species, the major chemical compounds were formaldehyde, acetone, and acetaldehyde [35].

### 3.4. Volatile Organic Compound Emissions during the Meat Grilling

VOCs emitted from cooking activities are important contaminant sources for the environmental quality, which are influenced by the food materials, cooking temperature, cooking methods, and oil types [13]. The individual concentrations of the 22 representative VOCs emitted from the meat-grilling, pre-marinated or not, are shown in Table 6. The total of 22 VOCs determined from the fumes could be divided into 7 types: aromatic hydrocarbons (ethylbenzene, 1,4-diethylbenzene, toluene, etc.), alkanes (n-hexane, n-heptane, etc.), ketones (3-pentanone, cyclopentanone, 2-heptanone, etc.), alkenes (1-decene, 1-dodecene, styrene, etc.), esters (ethyl acetate, n-butyl acetate, ethyl lactate, etc.), ethers (hexamethyldisiloxane, anisole), and alcohol (isopropyl alcohol) (Figure 4). The total concentration of the VOCs emitted from the grilling meat pre-marinated with salt, white pepper, garlic powder, and MS were 255.63 ± 85.63, 634.79 ± 121.68, 814.33 ± 201.67, and 2870.24 ± 403.35 μg/m^3^, respectively. Although the total VOCs of grilling meat pre-marinated with salt showed an ability for reducing emissions compared with the control group (476.67 ± 110 μg/m^3^), there was no statistically significant difference observed (*p* > 0.05). Among the other three pre-curing treatments, only the marinated group significantly increased the emissions concentration of total volatile organic compounds (*p* < 0.05). Spice additives (except salt) contain some natural antioxidant substances [39]. These substances will also undergo complex chemical reactions like fatty acids and protein in meat due to high temperatures in the grilling [1]. Generally, saturated fatty acids are more resistant to oxidation than unsaturated ones. Although the content of saturated fatty acids in pork is higher than unsaturated fatty acids, they can also be oxidized to produce cleavage products when the temperature of thermal processing exceeds 150 °C [40].

As shown in Figure 4, the aromatic hydrocarbons, ketones, and esters were the most abundant constituents of VOCs in the cooking fumes during the grilling, which is similar to previous publications. Lee et al. [28] investigated the emissions characteristics of VOCs from the grilling of ribs. Out of a total number of 88 compounds detected from the fumes, benzene and vinyl acetate were the dominant VOCs during the grilling. These results indicated that aromatic hydrocarbons were most of the VOCs in the cooking fumes for the process of meat-grilling. 

Previous studies also discussed the influence of different cooking methods on fume emission. Cheng et al. determined 51 volatile organic compound species from different cooking methods and proved that the total VOCs concentration produced by the grilling was the highest among the several cooking methods. One cause could be the grill being heated by charcoal which could generate more VOCs in the process of combustion [41]. In addition, the VOC formation was accompanied by the generation of oxidative radicals such as O_2_^−^ and OH during thermal processing. The existence of water can enhance the activity of oxidative radicals and promote chemical reactions with fatty acids to generate VOCs [42,43].

## 4. Conclusions

The marinated meat with white pepper, salt, garlic powder, or mixed spices can significantly reduce mass concentration of PM emissions in cooking fumes, especially the white pepper. PM_2.5_ accounts for a higher proportion of the total PM concentration. During the grilling process, formaldehyde, acetaldehyde, propionaldehyde, and acetone are quantitatively abundant carbonyl compounds. The low molecular weight carbonyl compounds (C1–C3) in cooking fumes are dominant in the grilling of marinated meat. Only the application of compound additives can significantly increase the total concentration of VOCs. The aromatic hydrocarbons are the predominant VOCs species, followed by ketone compounds. These results give an insight into the emission characteristics of the PM and VOCs from the grilling of meat marinated with additives and can be useful in the design of management strategies for controlling the fume emission and alleviating the pressure of ultimate purification.

## Figures and Tables

**Figure 1 foods-11-00833-f001:**
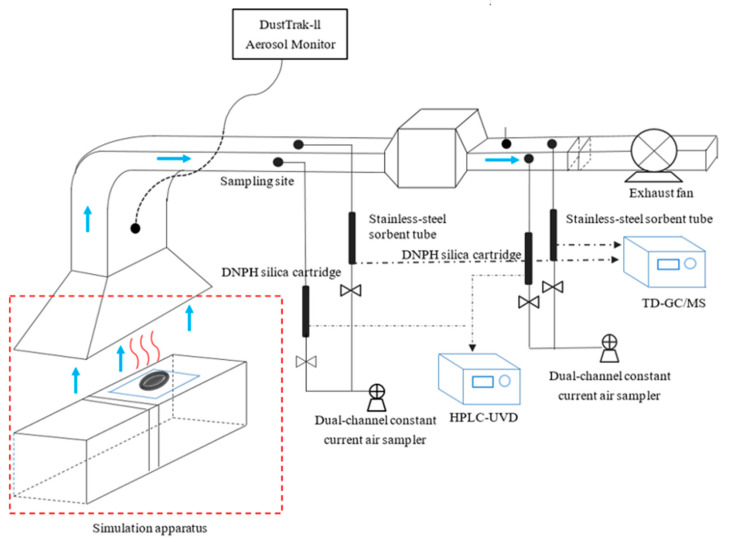
Schematic diagram of the test facility and grilling experiments.

**Figure 2 foods-11-00833-f002:**
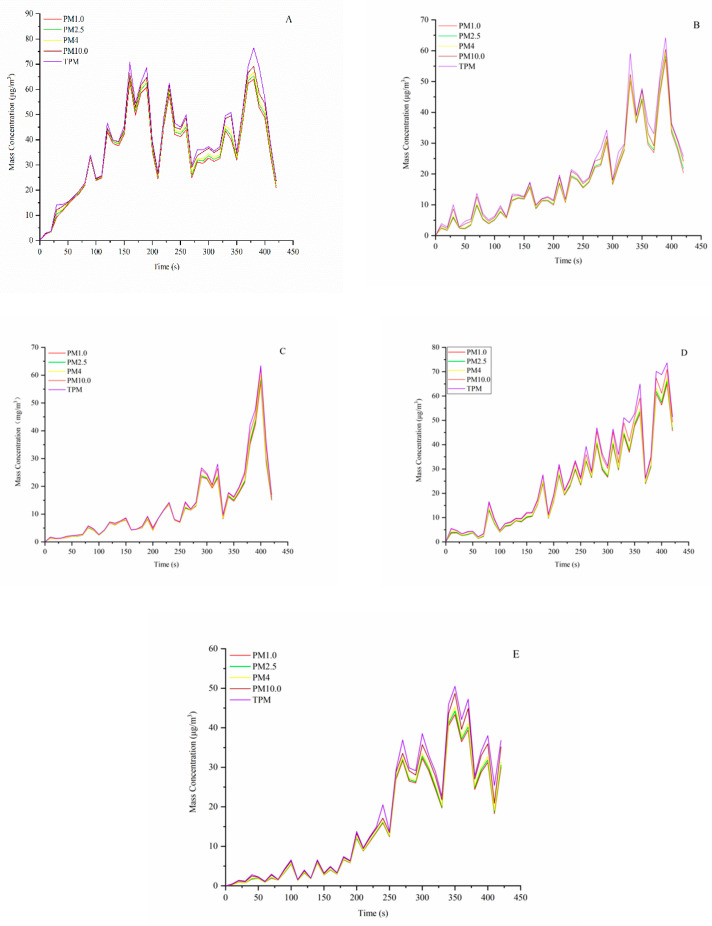
Measured time-dependent mass concentration of grilling-generated PM_1.0_, PM_2.5_, PM_4_, PM_10_, and TPM for four types of additives applied. The letters in the graph represent different experimental groups. (**A**) Control, (**B**) Salt group, (**C**) White pepper group, (**D**) Garlic powder group, and (**E**) Mixed spices group.

**Figure 3 foods-11-00833-f003:**
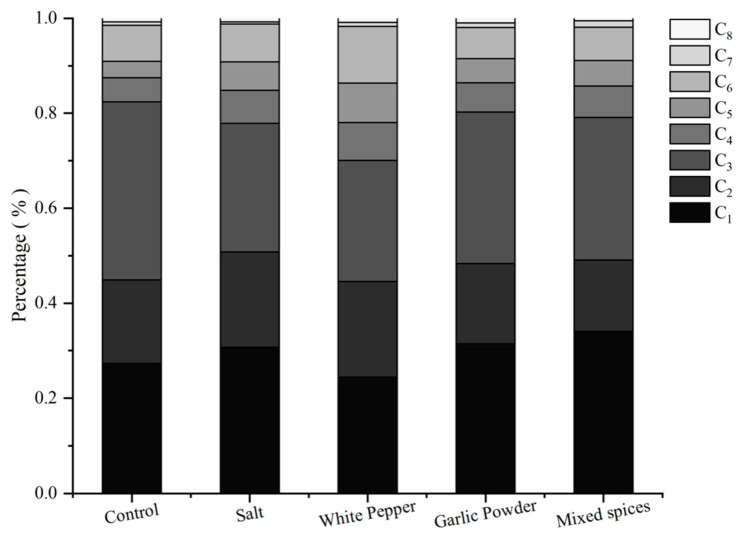
The relative proportions of C_1–8_ aldehyde and ketone compounds emitted from the grilling.

**Figure 4 foods-11-00833-f004:**
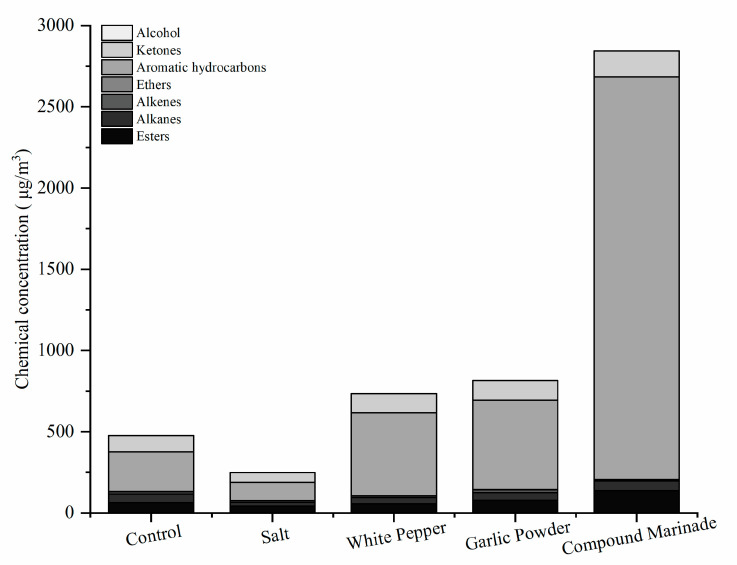
The speciation of VOCs detected in the cooking fumes during the meat grilling.

**Table 1 foods-11-00833-t001:** High Performance Liquid Chromatography gradient elution program (Time refers to elution time, A refers to acetonitrile, B refers to ultrapure water).

Time (min)	A (%)	B (%)
0	0	0
20	60	40
30	100	0
32	60	40
40	60	40

**Table 2 foods-11-00833-t002:** Molecular formula and physical properties of target chemicals.

Compound	MF	RMW	BP (°C)	CAS NO
Formaldehyde	CH_2_O	30.03	−19	50-00-0
Acetaldehyde	C_2_H_4_O	44.05	21	75-07-0
Acrolein	C_3_H_4_O	56.06	52.5	107-02-8
Acetone	C_3_H_6_O	58.08	56.53	67-64-1
Propionaldehyde	C_3_H_6_O	58.08	47.9	123-38-6
Crotonaldehyde	C_4_H_6_O	70.09	102.2	123-73-9
Methacrolein	C_4_H_6_O	70.09	69	78-85-3
2-Butanone	C_4_H_8_O	72.11	79.6	78-93-3
Butyraldehyde	C_4_H_8_O	72.11	77.6	123-72-8
Benzaldehyde	C_7_H_6_O	106.12	179	100-52-7
Valeraldehyde	C_5_H_10_O	86.13	103.7	110-62-3
*m*-Tolualdehyde	C_8_H_8_O	120.15	199	620-23-5
*n*-Hexaldehyde	C_6_H_12_O	100.16	130–131	66-25-1

The MF refers to the molecular formula; The RMW refers to the relative molecular mass; The BP refers to the boiling point of the compounds.

**Table 3 foods-11-00833-t003:** Mass concentration and size distribution of PM_1.0_, PM_2.5_, PM_4.0_, and PM_10_ emitted from the cooking fumes.

Additive Types	Mass Concentration			
	PM_1.0_	PM_2.5_	PM_4.0_	PM_10_
Control	36.20 ± 5.28 ^a^	37.06 ± 5.45 ^a^	37.73 ± 5.54 ^a^	39.03 ± 5.95 ^a^
Salt	18.33 ± 3.18 ^bc^	18.63 ± 3.20 ^bc^	19.00 ± 4.12 ^bc^	20.06 ± 2.93 ^bc^
White pepper	12.50 ± 4.48 ^d^	12.78 ± 4.13 ^d^	12.96 ± 5.16 ^d^	13.73 ± 4.14 ^d^
Garlic powder	23.13 ± 2.65 ^b^	23.50 ± 2.30 ^b^	24.03 ± 3.02 ^b^	25.87 ± 3.32 ^b^
MS	15.20 ± 0.43 ^cd^	15.47 ± 0.97 ^cd^	15.73 ± 0.49 ^cd^	16.87 ± 1.36 ^cd^

Mass concentration unit: mg/m^3^. Values bearing different lowercase letters in the same column are significant differences (*p* < 0.05).

**Table 4 foods-11-00833-t004:** The percentage of PM_2.5_ to total particles concentration emitted from the meat-grilling.

Group	Mass Concentration (mg/m^3^)		The Percentage of PM_2.5_/TPM
	PM_2.5_	TPM	
Control	37.07 ± 5.45	40.47 ± 5.16	91.59%
Salt	18.63 ± 3.23	21.10 ± 3.52	88.29%
White pepper	12.77 ± 4.51	14.13 ± 4.09	90.38%
Garlic powder	23.50 ± 2.69	27.17 ± 2.97	86.49%
MS	15.47 ± 0.46	17.8 ± 0.95	86.91%

**Table 5 foods-11-00833-t005:** The mean concentration of 13 target carbonyl compounds and their relative proportions distribution.

Carbon Number	Compounds	Control		Salt		White Pepper		Garlic Powder		MS	
		Concentration	Percentage	Concentration	Percentage	Concentration	Percentage	Concentration	Percentage	Concentration	Percentage
C1	Formaldehyde	2277.15 ± 188.82 ^b^	27.33%	2308.96 ± 124.57 ^b^	30.69%	1633.59 ± 122.97 ^a^	24.40%	2483.52 ± 132.96 ^b^	31.44%	2818.39 ± 171.27 ^c^	34.01%
C2	Acetaldehyde	1465.1 ± 303.15 ^a^	17.59%	1510.59 ± 167.01 ^a^	20.08%	1347.6 ± 135.20 ^a^	20.13%	1333.40 ± 106.21 ^a^	16.88%	1250.66 ± 218.59 ^a^	15.09%
C3	Acrolein	ND	0	ND	0	ND	0	ND	0	ND	0
C3	Acetone	2320.76 ± 356.41 ^b^	27.86%	1408.51 ± 179.49 ^a^	18.72%	1127.81 ± 119.85 ^a^	16.85%	1924.84 ± 135.58 ^b^	24.37%	1992.17 ± 167.96 ^b^	24.04%
C3	Propionaldehyde	799.39 ± 166.56 ^b^	9.60%	631.76 ± 76.39 ^a^	8.39%	581.23 ± 60.55 ^a^	8.69%	595.72 ± 74.73 ^a^	7.54%	497.22 ± 99.80 ^a^	6.00%
C4/Linear	Crotonaldehyde	56.85 ± 2.75 ^a^	0.68%	45.44 ± 2.36 ^a^	0.60%	56.89 ± 8.29 ^a^	0.85%	49.04 ± 3.30 ^a^	0.62%	56.43 ± 4.38 ^a^	0.68%
C4/Linear	Butyraldehyde	306.28 ± 19.98 ^a^	3.67%	401.55 ± 13.37 ^b^	5.33%	396.18 ± 27.07 ^b^	5.92%	363.62 ± 0.95 ^b^	4.60%	385.25 ± 8.8 ^b^	4.65%
C4/Branch	Methacrylaldehyde	24.73 ± 2.53 ^a^	0.29%	21.56 ± 6.76 ^a^	0.29%	23.55 ± 6.61 ^a^	0.35%	37.54 ± 2.77 ^b^	0.48%	54.66 ± 2.66 ^c^	0.66%
C4/Linear	2-Butanone	39.99 ± 4.41 ^a^	0.48%	55.07 ± 4.56 ^b^	0.73%	57.76 ± 2.86 ^b^	0.86%	39.06 ± 19.07 ^a^	0.49%	49.06 ± 5.87 ^ab^	0.59%
C5/Linear	Valeraldehyde	287.08 ± 34.23 ^a^	3.45%	450.13 ± 63.90 ^b^	5.98%	557.90 ± 20.61 ^c^	8.33%	400.43 ± 30.79 ^b^	5.07%	447.98 ± 14.62 ^b^	5.41%
C6/Linear	n-Hexaldehyde	629.28 ± 27.24 ^b^	7.55%	599.40 ± 93.31 ^b^	7.97%	796.73 ± 19.26 ^c^	11.90%	520.06 ± 14.61 ^a^	6.58%	580.23 ± 37.62 ^ab^	7.00%
C7/Ring	Benzaldehyde	63.85 ± 12.25 ^a^	0.77%	34.78 ± 24.07 ^a^	0.46%	55.63 ± 0.36 ^a^	0.83%	76.77 ± 7.36 ^ab^	0.97%	111.44 ± 4.50 ^b^	1.35%
C8/Ring	m-Tolualdehyde	60.86 ± 2.45 ^ab^	0.73%	56.84 ± 19.60 ^a^	0.76%	59.50 ± 5.61 ^ab^	0.89%	76.02 ± 6.11 ^c^	0.96%	43.18 ± 2.96 ^a^	0.52%
Total concentration		8331.33 ± 274.44 ^a^		7524.59 ± 218.16 ^b^		6694.37 ± 456.25 ^c^		7900.02 ± 74.43 ^ab^		8286.645 ± 269.55 ^a^	

Carbonyl compounds concentration unit: μg/m^3^. Values bearing different lowercase letters in the same line are significant differences (*p* < 0.05). ND stands for no detection of this compound.

**Table 6 foods-11-00833-t006:** VOC concentrations in the cooking fumes emitted from the grilling of meat marinated with different additives.

Chemical Compound	Chemical Concentration (μg/m^3^)				
	Control	Salt	White Pepper	Garlic Powder	MS
Isopropyl alcohol	ND	ND	ND	ND	ND
n-Hexane	13.23 ± 8.03 ^a^	5.30 ± 2.83 ^a^	6.33 ± 3.93 ^a^	9.33 ± 1.6 ^a^	9.47 ± 5.06 ^a^
Ethyl acetate	7.87 ± 3.93 ^a^	5.50 ± 1.70 ^a^	3.13 ± 2.8 ^a^	15.13 ± 10.07 ^a^	54.23 ± 6.78 ^b^
Hexamethyldisiloxane	ND	ND	ND	ND	ND
Benzene	192.5 ± 12.9 ^b^	73.53 ± 28.87 ^a^	405.13 ± 89.6 ^c^	516.07 ± 125.6 ^c^	2251.3 ± 260.44 ^d^
n-Heptane	40.03 ± 6.77 ^a^	17.13 ± 10.53 ^a^	30.3 ± 18.17 ^a^	38.10 ± 4.43 ^a^	51.23 ± 26.70 ^a^
3-Pentanone	39.67 ± 17.07 ^ab^	28.70 ± 10.90 ^a^	41 ± 4.07 ^ab^	43.20 ± 3.27 ^ab^	59.17 ± 2.97 ^b^
Toluene	12.2 ± 2.43 ^b^	4.47 ± 1.07 ^a^	30.7 ± 6.3 ^c^	10.06 ± 2.06 ^b^	162.53 ± 21.39 ^d^
n-Butyl acetate	44.8 ± 13.93 ^ab^	27.4 ± 8.13 ^a^	45.33 ± 5.93 ^ab^	53.33 ± 8.00 ^ab^	66.83 ± 5.03 ^b^
Cyclopentanone	12.33 ± 11.26 ^a^	7.00 ± 4.73 ^a^	29.9 ± 6.83 ^b^	25.33 ± 2.07 ^b^	33.27 ± 2.34 ^b2^
Ethyl lactate	ND	0.47 ± 0.08	0.13 ± 0.13	ND	ND
Ethylbenzene	8.1 ± 2.17 ^a^	7.03 ± 1.50 ^a^	7.53 ± 0.4 ^a^	8.80 ± 1.60 ^a^	15.02 ± 1.45 ^b^
1,4-Diethylbenzene	14.23 ± 4.63 ^a^	16.73 ± 2.40 ^ab^	11.37 ± 1.23 ^a^	11.40 ± 1.53 ^a^	25.4 ± 2.50 ^b^
1,3-Xylene	13.33 ± 4.13 ^a^	15.43 ± 2.17 ^ab^	10.53 ± 1.13 ^a^	10.50 ± 1.43 ^a^	23.53 ± 2.23 ^b^
propylene glycol methyl ether	11.2 ± 2.08 ^a^	6.17 ± 2.7 ^a^	9.4 ± 2.00 ^a^	9.03 ± 2.63 ^a^	15.17 ± 0.84 ^b^
1,2-Dimethylbenzene	4.96 ± 1.43 ^a^	5.3 ± 0.97 ^a^	4.23 ± 0.23 ^a^	2.70 ± 2.63 ^a^	12.17 ± 1.00 ^b^
Styrene	8.47 ± 1.40 ^a^	5.33 ± 2.07 ^a^	8.20 ± 1.27 ^a^	10.90 ± 2.50 ^a^	8.13 ± 6.59 ^a^
2-Heptanone	33.63 ± 5.30 ^b^	17.40 ± 7.13 ^a^	28.43 ± 2.17 ^a^	32.90 ± 3.43 ^b^	48.33 ± 4.95 ^c^
Anisole	0.3 ± 0.03 ^a^	0.77 ± 0.17 ^a^	0.67 ± 0.13 ^a^	1.77 ± 0.10 ^b^	2.53 ± 0.38 ^b^
1-Decene	2.5 ± 0.3 ^b^	3.30 ± 0.33 ^b^	1.8 ± 0.05 ^a^	4.07 ± 2.27 ^b^	9.733 ± 0.49 ^c^
1-Dodecene	2.73 ± 0.33 ^a^	1.37 ± 0.70 ^a^	1.93 ± 0.33 ^a^	2.07 ± 0.07 ^a^	4.56 ± 0.29 ^b^
2-Nonanone	14.57 ± 2.17 ^a^	7.30 ± 2.77 ^a^	18.73 ± 3.20 ^b^	19.63 ± 12.76 ^b^	17.63 ± 1.77 ^b^
Total concentration	476.67 ± 110 ^ab^	255.63 ± 85.63 ^a^	634.79 ± 121.68 ^b^	814.33 ± 201.67 ^b^	2870.24 ± 403.35 ^c^

VOCs concentration unit: μg/m^3^. Values bearing different lowercase letters in the same line are significant differences (*p* < 0.05). ND stands for no detection of this compound.

## Data Availability

Data is contained within the article or Appendix A.

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
