# Peer review of "Assessing Impacts of Additives on Particulate Matter and Volatile Organic Compounds Produced from the Grilling of Meat"

_foods, 2022, doi:10.3390/foods11060833_

Round 1
Reviewer 1 Report
Please see the attached file.

Author Response
Response to Editor/reviewer 1#
Thank you so much for your comments and suggestions. Our manuscript has been revised based on your comments. The details are shown as following.
Responds to the reviewer’s comments:
Reviewer #1:
- Reviewer’s comment: Page2, line 6 – Previous.
Response: Thanks for your question and good suggestion. There is a misspelled word on our manuscript and our manuscript was revised based on your comments.
- Reviewer’s comment: Page 2, paragraph 1 where author talk about the cardiovascular and respiratory impact of cooking fumes, it is also good to discuss the potential nervous impact of cooking fumes. The following article can be useful: the impact of frying aerosol on human brain activity, https://doi.org/10.1016/j.neuro.2019.06.008.
Response: Thanks for your advice. We have read this article carefully and will discuss the potential nervous impact of cooking fumes from the perspective of this article. Our manuscript was revised based on your comments. The details were shown as Lines:
And a number of studies have proved that PM and VOCs emitted from cooking activities consisted of multiple hazardous chemical compounds which are easily deposited in the alveoli and can cause cardiovascular and respiratory disease or even to death. Apart from these negative effects, the cooking ultrafine particles can also impact on human brain activity [4-9]. Therefore, characterizing VOCs, PM and its chemical constituent and exploring beneficial method to reduce the emission of cooking fumes are significant in improving healthy environment of food processing or cooking for human beings.
- Reviewer’s comment: Page 2, paragraph 2,reference 11 never discussed the statements in the manuscript. Apparently, the authors cited to the wrong article, by mistake. Please correct it.
Response: Thanks for your carefully checking. We have carefully checked the manuscript and corrected the reference numbering.
- Reviewer’s comment: Page 2, paragraph 3, cites to Torkmahalleh et al. as reference 12, but this in the reference list, reference 12 is reserved for Peng et al.
Response: Thanks for your suggestions. Our manuscript was revised based on your comments.
- Reviewer’s comment: I believe there is a mistake in reference numbering in the reference list or in the text. Please fix it.
Response: Thanks for your carefully checking. We have carefully checked the manuscript and corrected the reference numbering.
- Peng, C.-Y.; Lan, C.-H.; Lin, P.-C.; Kuo, Y.-C. Effects of cooking method, cooking oil, and food type on aldehyde emissions in the cooking oil fumes. J. Hazard. Mater. 2017, 324, 160-167.
- Amouei Torkmahalleh, M.; Gorjinezhad, S.; Keles, M.; Unluevcek, H.S.; Azgin, C.; Cihan, E.; Tanis, B.; Soy, N.; Ozaslan, N.; Ozturk, F.; Hopke, P.K. A controlled study for the characterization of PM2.5 emitted during grilling ground beef meat. J. Aerosol. Sci. 2017, 103, 132-140.
- Reviewer’s comment: In section 2.3 no information about temperature.
Response: Thanks for your comment. We have revised the references of the manuscript according to your comment.
- Reviewer’s comment: According to Figure 1, please clarify that the samples taken after the quartz filter which are introduced to GC-MS and HPLC are free of particles or not. Other words are the data analyzed by HPLC and GC-MS are for only gas emissions or gas and particles?
Response: Thanks for your questions. The particulate matter samples collected onto quartz fiber filters are not directly analyzed by instrument. After extraction and purification, it was analyzed and determined by GC-MS. Procedures of sample extraction and purification are as follows. Particulate samples were ultrasonically extracted three times, each with 15 mL mixture of hexane, dichloromethane and acetone (2:2:1 in volume). The combined extract was dried with anhydrous sodium sulfate and purified on a neutral silica gel column with hexane as the eluent. The collected extract was further concentrated to 50 μL in a vial under a gentle stream of N2 with the internal standards before instrumental analysis. And the original purpose of sampling PM emitted from cooking fumes onto quartz fiber filters is to characterize the emission of polycyclic aromatic hydrocarbons on PM. We do not intend to put these relevant experimental data in this article and failed to update figure 1 in time when writing this article. This is our deficiency, and our manuscript was revised based on your comments.
- Reviewer’s comment: It is not clear how many times the experiments were repeated for control and cooking.
Response: Thanks for your questions. Our manuscript was revised based on your comments. The details were shown as section 2.7:
All the experiments in this study were performed in triplicates.
- Reviewer’s comment: Experiments in section 3.1, the authors discuss the order in mass concentration to be PM10 > PM4 > PM2.5 > PM1 and then cite it to reference 21. This order is quite obvious and expectable as in definition PM10 contains PM4, PM2.5, and PM1 and likewise PM4 contains PM2.5 and PM1. Thus, always this order has to be achieved. Otherwise, something is wrong with the instrument. I suggest to delete this discussion from the manuscript.
Response: Thanks for your questions. We have deleted this discussion in the manuscript.
- 1 Reviewer’s comment:Figure 3 does not really provide new information as figure 2 already covers the information in figure3. Please move it to the supplementary materials.
Response: Thanks for your advice. Our manuscript was revised based on your comments and we have already moved figure 3 into the supplementary data.
- 1 Reviewer’s comment:Given the grilling period which was around 450 seconds, please explain what was the level of the grilling (medium or well-done).
Response: Thanks for your comment. The level of grilling was related to the time. According to the data obtained from our pre-experiment, the internal center temperature of meat can reach 85℃ when the time of grilling is about 420 seconds. Therefore, the time of grilling in our formal experiment was 420 seconds. At this grilling time, the meat products were already well-done.
12 Reviewer’s comment: Apparently, the DustTrak was not calibrated against gravimetric mass measurements. DustTrak principle of working is light scattering which typically overestimates mass concentration. Thus, a side by side comparison between this device and gravimetric mass measurements using is needed to correct the concentrations. Author should create a section named as “ Limitation of this study” and explain the lack of this data in their manuscript.
Response: Thanks for your advice. Our manuscript was revised based on your comments.
13 Reviewer’s comment: It objective of presenting figure 4 is not very well clear? What point the authors want to raise by this figure. I suggest they might be able to discuss it from exposure point. For example, people when they do grilling, they are present at the beginning of the grilling to adjust the position of the meat etc or two roll it over and then they may leave the grilling facility until it cooks itself. Thus, it is important to reduce the exposure at the early stages of the grilling when the chef is present over the stove. In this case, Figure 4 shows that marinating the meat will reduce the PM concentration significantly at the early stage of the grilling.
Response: Thanks for your advice. Our manuscript was revised based on your comments and perspective.
14 Reviewer’s comment: The English of the manuscript needs to be improved significantly.
Response: Thanks for your advice. I will try my best to improve my English writing ability.
- 1 Reviewer’s comment: In Tables 4 and 5, it is not clear which pairs are statistically significant and which are not. The lower letter cases are confusing. Please discuss the statistically significant pairs, only between control and marinated ones (no need to discuss changes among the additives), in the main text and clarify which scenarios were statistically significant for the total carbonyls/VOCs and individual carbonyls/VOCs. Thus, the rest, which are not discussed, should be considered as insignificant. Then, from this point conclude if the additives had an impact on the total and individual carbonyl and VOCs or not.
Response: Thanks for your questions. Our manuscript was revised based on your comments.
- 1 Reviewer’s comment: This study showed that additives particularly salt can reduce PM. Please discuss the reason beyond this observation. For example, PM should be the results of the nucleation of the low volatile organic compounds emitted during grilling for example high carbon carbonyl, etc. Can we conclude that additives reduced the emission of these types of compounds in the gas phase (as partly showed in Table 4 for salt ) and thus the nucleation rate of these compounds decreased during grilling and PM emission reduced? Basically the mechanisms for the impact of additives, particularly salts on the PM emissions are not discussed in this manuscript. The author might find the following article useful although it is related to the heating cooking oils
Response: Thanks for your questions and advice. Our manuscript was revised based on your comments and perspective.
- 1 Reviewer’s comment: Was any analyses conducted on the PM samples collected by the quartz filters? What was the purpose of collecting PM on the quartz filters? Only separating particles from gas phase before analyzing by GC-MS and HPLC? If yes, then please clearly state it in the manuscript. If this filter sampling was aimed for further analyses, then provide the data.
Response: Thanks for your questions. And the original purpose of sampling PM emitted from cooking fumes onto quartz fiber filters is to characterize the emission of polycyclic aromatic hydrocarbons on PM. We do not intend to put these relevant experimental data in this article.
- 1 Reviewer’s comment:Following the previous comments, the authors showed that the emissions were analyzed before and after PM collection on the quartz filters. This means the GC and HPLC results should be presented before and after PM collections. Other words, the authors should show what were the VOCs and carbonyl contents of the emissions for (gas + PM) and (only gas). Then they can conclude the VOCs and carbonyl content of only PM. They can then discuss the impact of additives on the PM carbonyl and VOCs (total and individual) as well and compare the results with the control and discuss the statistically significant cases. However, this requires the assumption that quartz filter remove the totally the PM before the emissions being introduced to the GC and HPLC.
Response: Thanks for your questions. This is our mistake in revising the manuscript. According to the research articles we have investigated, carbonyl compounds and volatile organic compounds in cooking fumes mainly exit in gaseous form. Carbonyl compounds and volatile organic compounds in cooking fumes are not adsorbed on particulate matter. Therefore, the design scheme of our experiment only analyzed gaseous carbonyl compounds and volatile organic compounds in cooking fumes.
- 1 Reviewer’s comment:The authors did not discuss why additives (except salt) showed higher VOCs compared to the control case although only one case showed statistically significant difference. Canwe conclude additives (except salt) examined in this study themselves contribute to the VOCs and Carbonyl? As a results when additives were used, VOCs increased? Please discuss and cite properly, the VOCs and possibly carbonyl emissions from the heated additives.
Response: Thanks for your questions. This is our deficiency in data discussion and analysis. We will modify the discussion content according to your suggestions
- 2 Reviewer’s comment:Please discuss the results of your study with the following reference that showed addition of salt to the meat may increase the PM emissions.
(O'Leary, C., de Kluizenaar, Y., Jacobs, P., Borsboom, W., Hall, I., & Jones, B. (2019). Investigating measurements of fine particle (PM 2.5) emissions from the cooking of meals and mitigating exposure using a cooker hood. Indoor air, 29(3), 423-438.)
Response: Thanks for your questions and advice. We have read this article carefully and will discuss the potential nervous impact of cooking fumes from the perspective of this article. Our manuscript was revised based on your comments.
- 2 Reviewer’s comment: The last name of the author “Torkmahalleh” in the text and reference list has to be changed to the correct last name “Amouei Torkmahalleh”. Thus, it should be cited as Amouei Torkmahalleh et al. and listed properly in the reference list.
Response: Thanks for your carefully checking. We have carefully checked the last name of authors in the manuscript and corrected it.

Reviewer 2 Report
The manuscript is about the effect of various additives on the formation of particulates and volatile organic compounds (VOCs) in grilled pork meat. It is an interesting study, although I have doubts about the justification that there is no investigation similar to it. The description of the experiment is weak, beginning from the used materials. For example, what is a "compound marinade"? It is commercial, but it would be important to know at least the compositioin fo this marinade. Experimental conditions are not fully described, for eample, at the collection of samples, which is the speed drive, velocity, tie, flow rate? It is very generally described.
What could be "sousing"?
In some place in the text, the authors describe "low temperature"- which temperature it would bem?
Which is the origin of HDBC used and why it was used<
The authors describe that moisture and lipid content of each batch is the same? Where are the results? How nay batches there would be? Which methodology was used for measuring these parameters?
Item 2.3. For how long was the monitoring of the particles? A better description is requiresd.
Item 2.6 How the VOCs were identified? By MS, but describe the methodology. for quantifiction, standrads were used? Note that for using the area, it is meant that all of the VOCs are in all treatments. Is it true?
In page 9, there is mention of chicken sking. The experiment was with pork.
Figure 2 - statistical difference of the treatments?
Figure 4 - legend incomplete.
In some places of the manuscript the authors mention "prpevioes publication", but there is no reference or it is not clear that is them following described publication.
Author Response
Response to Editor/reviewer 2#
Thank you so much for your comments and suggestions. Our manuscript has been revised based on your comments. The details are shown as following.
Responds to the reviewer’s comments:
Reviewer #2:
1 Reviewer’s comment: Experimental conditions are not fully described, for example, at the collection of samples, which is the speed drive, velocity, tie, flow rate? It is very generally described.
Response: Thanks for your recommendation. Our manuscript was revised based on your comments.
- Reviewer’s comment:What could be “sousing”?
Response: Thanks for your questions. The meaning of this word is same as marinade. This is our deficiency in revising the manuscript. Our manuscript was revised based on your comments.
- Reviewer’s comment:In some place in the text, the authors describe “low temperature” which temperature it would bem?
Response: Thanks for your questions. When we used the additives to marinade the meat, we stored the pre-marinade meat in the cold storage in our laboratory, and the ambient temperature maintain 4 ℃.
- Reviewer’s comment:Whichis the origin of HDBC used and why it was used.
Response: Thanks for your comment. The full name of HDBC is high density bamboo charcoal, and it is made up of bamboo powder by secondary pressing. Compared with the traditional fruit charcoal, the density of bamboo charcoal produced by secondary extrusion is higher.
In the previous experiments, we have investigated the emission characteristics of cooking fumes when high density bamboo charcoal and natural fruit charcoal were used for grilling, we found that the mass concentration of particulate matter emitted from grilling was lower when high density bamboo charcoal was used as fuel. Therefore, we selected high density bamboo charcoal to grill.
- Reviewer’s comment:The authors describe that moisture and lipid content of each batch is the same? Where are the results? How nay batches there would be? Which methodology was used for measuring these parameters?
Response: Thanks for your recommendation. Unfortunately, we failed to consider the determination of moisture and lipid content in the design of the experimental scheme, which was the deficiency of our experimental scheme.
- Reviewer’s comment:Item 2.3. For how long was the monitoring of the particles? A better description is required.
Response: Thanks for your recommendation. This manuscript has been revised based on your comments. The details were shown as follows:
The whole processing time of grilling was 420secs. The PM mass concentration, size distribution of PM less than 1.0 μm (PM1.0), PM less than 2.5 μm (PM2.5), PM less than 4.0 μm (PM4.0), PM less than 10.0 μm (PM10), and total concentration were measured using a DustTrak Aerosol Monitor (8533, TSI, USA) equipped with an electrostatic prevention hose.
- Reviewer’s comment:Item 2.6 How the VOCs were identified? By MS, but describe the methodology. for quantification, standards were used? Note that for using the area, it is meant that all the VOCs are in all treatments. Is it true?
Response: Thanks for your comment. In order to characterize the compounds in the sampling tube, the retention time of the compounds was compared with the mass spectrum library. As for the quantification of VOCs, we used the mixed standard of 22 compounds dissolved in methanol to make the calibration curve. Then, we used the calibration curve equation to calculate the compound concentration.
- Reviewer’s comment:In page 9, there is mention of chicken skin. The experiment was with pork.
Response: Thanks for your comment. The raw material of the experiment in this research is pork, and the word “chicken skin” in the manuscript was an error when we revised the manuscript. Our manuscript was revised based on your comments.
- Reviewer’s comment:Figure 2 - statistical difference of the treatments?
Response: Thanks for your recommendation. our manuscript was revised based on your comments. The details of figure 2 were shown in revised manuscripts:
- Reviewer’s comment:Figure 4 - legend incomplete.
Response: Thanks for your comment. The legend of figure 4 in this manuscript has been revised based on your comments. The details were shown as follows:
Figure 4. Measured time-dependent mass concentration of grilling-generated PM1.0, PM2.5, PM4, PM10, and TPM for 4 types of additives applied. The letters in the graph represent different experimental groups. (A) Control, (B) Salt group, (C) White pepper group, (D) Garlic powder group, and (E) Compound marinade group.
- 1 Reviewer’s comment:In some places of the manuscript the authors mention "previous publication", but there is no reference or it is not clear that is the following described publication.
Response: Thanks for our comment. We have revised the references of the manuscript according to your comment.

Reviewer 3 Report
I think that the topic is very interesting for the journal, since it focuses on the evaluation of the effect of four additives commonly used in grilling meat on volatile organic compounds (VOCs), particulate matter and carbonyl compounds of pork belly. This topic is important since several chemical reactions occurred during cooking that could change meat final appearance, flavour and nutrient contents meaning that they modify its quality and consumer acceptability. However, this work needs to be reviewed in depth. I recommend a major revision.
General comments
- Put the manuscript in the journal format. It is difficult to do the revision when line numbers are missing.
- I miss the determination of the chemical composition of each of the batches studied, since it would allow to justify some of the results found.
- Why were protein degradation compounds determined and not lipid degradation compounds?
Specific comments
Abstract
- The term "compound marinade" is not used correctly since it seems to make no sense. Please rewrite it correctly or use a term that better defines this batch. Use it throughout the text.
- Define "TPM" the first time it appears in the text.
Introduction
- Reflect in the introduction that chemical reactions occurred during cooking could change meat final quality and consumer acceptability.
- What do you mean by “effectively alleviating the pressure of ultimate purification?
Material and methods
- Combine Sections 2.1. and 2.3. in a single one that has as a title for example “Materials, chemicals, and sampling instruments”.
- Figure 1 is very explanatory since it allows a clear understanding of how the processing of the samples was carried out. However, Section 2.2. should begin by showing which batches were used in the study, including the information showed below about the control batch (“As a control, the same lean pork was conducted without any spice additive”).
- Did you do a pre-treatment of the data (normal distribution and variance homogeneity)?
Results and Discussion
- Discuss the results using more recent references.
Author Response
Response to Editor/reviewer 3#
Thank you so much for your comments and suggestions. Our manuscript has been revised based on your comments. The details are shown as following.
Responds to the reviewer’s comments:
Reviewer #3:
- 1. Reviewer’s comment: Put the manuscript in the journal format. It is difficult to do the revision when line numbers are missing.
Response: Thanks for your comment. Unfortunately, there were no line numbers in the manuscript with the journal format. We have edited the line numbers in the manuscript before we submitted the manuscript in the journal system. This was probably because the editor forgot to edit the number of lines when the manuscript was in typesetting.
- Reviewer’s comment:Why were protein degradation compounds determined and not lipid degradation compounds?
Response: Thanks for your comment. Various complex reactions will occur in the thermal processing of grilling. Each volatile organic compound was produced in different pathways during thermal processing. Protein degradation compounds and lipid degradation compounds were determined and the chemical compounds emitted form cooking fumes were the mixture generated from different reactions. We discussed the date from on of the perspectives, which may be our shortcomings.
- Reviewer’s comment:The term "compound marinade" is not used correctly since it seems to make no sense. Please rewrite it correctly or use a term that better defines this batch. Use it throughout the text.
Response: Thanks for your recommendation. We have revised the references of the manuscript according to your comment.
- Reviewer’s comment:Define "TPM" the first time it appears in the text.
Response: Thanks for your comment. The full name of “TPM” is total particles mass concentration. Our manuscript was revised based on your comments. The details were shown as follows:
Particle mass concentration and size distribution were monitored with a TSI Model 8533 DustTrak-DRX Aerosol Monitor (St, Paul, MN, USA), which has PM1.0, PM2.5, PM4.0, PM10, and TPM(total particles mass concentration) inlets.
- Reviewer’s comment:Reflect in the introduction that chemical reactions occurred during cooking could change meat final quality and consumer acceptability.
Response: Thanks for your recommendation. We have revised the manuscript according to your comment. The details were shown as follows:
Additives are widely applied in the different cooking methods to increase the taste and flavor. Meanwhile, the addition of various additives can also improve the quality of meat products and consumer acceptability. Black pepper, turmeric, salt, and garlic powder are commonly used additives in many cooking foods, especially in grilling food.
- Reviewer’s comment:What do you mean by “effectively alleviating the pressure of ultimate purification?
Response: Thanks for your comment. This was related to the research content of our subject, which aimed to research how to reduce the emission of cooking fumes (include the particulate matter and volatile organic compounds etc.) during the thermal processing of food. And in our research, we found that the addition of additives can reduce the emission of particulate matter emitted from the cooking fumes. That was why we said that “effectively alleviating the pressure of ultimate purification”.
- Reviewer’s comment:Combine Sections 2.1. and 2.3. in a single one that has as a title for example “Materials, chemicals, and sampling instruments”
Response: Thanks for your recommendation. Our manuscript was revised based on your comments. The details were shown as follows:
2.1 Materials, and sampling instruments
Four additives including salt, white pepper, garlic powder, and compound marinade are commonly used in the grilling of meat and were purchased from local supermarket in HeFei, P.R, China. Pork belly and high density bamboo charcoal (HDBC) also purchased from a local Carrefour supermarket (HeFei, P.R, China).
Particle mass concentration and size distribution were monitored with a TSI Model 8533 DustTrak-DRX Aerosol Monitor (St, Paul, MN, USA), which has PM1.0, PM2.5, PM4.0, PM10, and TPM(total particles mass concentration) inlets. Although the DustTrak Aerosol Monitor captures only a limited portion of PM mass concentration, the aerodynamic equivalent diameter of particulate matter greater than 500 nm constitutes the majority of the estimated PM mass concentration [15]. A JCH-2400 Dual-channel constant current air sampler (Qingdao Juchang Environmental Protection Group Co, Ltd, China) was also used as sampling pump. Quartz fiber filters (90 mm, Munktell, Sweden) were used to collect particulate matter.
- Reviewer’s comment:Figure 1 is very explanatory since it allows a clear understanding of how the processing of the samples was carried out. However, Section 2.2. should begin by showing which batches were used in the study, including the information showed below about the control batch (“As a control, the same lean pork was conducted without any spice additive”).
Response: Thanks for your recommendation. We have revised the manuscript according to your comment.
- Reviewer’s comment:Did you do a pre-treatment of the data (normal distribution and variance homogeneity)?
Response: Thanks for your comment. All our experimental conditions were analyzed on the basis of our pre-experiment. The temperature of grilling, the types of fuel nad the flow rate of sampling pump during the sampling were all determined by pre-experiment.
- 1 Reviewer’s comment:Discuss the results using more recent references.
Response: Thanks for your recommendation. We have revised the manuscript according to your recommendation and appropriately increase the number of references.

Round 2
Reviewer 1 Report
I have checked the author's responses to my comments and concerns. Surprisingly, most of my comments were not addressed. Even if the authors stated, " the manuscript has been improved based on your comments". I could not find changes to the manuscript based on what the authors claimed, except for some minor changes. Thus, I can say the majority of my comments were not addressed. For example, comments number 7, 9, 12, 13, 15, etc. Even authors discussed two papers cited as reference 10 and 13 but these papers do not really discuss what authors cited. At this point, my recommendation is to give another chance to the author to revise their manuscript. Maybe they uploaded the wrong file.
Author Response
Response to reviewer 1
Thank you so much for your comments and suggestions. Our manuscript has been revised based on your comments. The details are shown as following.
Responds to the reviewer’s comments:
Reviewer #1:
- Reviewer’s comment: Page2, line 6 – Previous.
Response: Thanks for your question and good suggestion. There is a misspelled word on our manuscript and our manuscript was revised based on your comments.
- Reviewer’s comment: Page 2, paragraph 1 where author talk about the cardiovascular and respiratory impact of cooking fumes, it is also good to discuss the potential nervous impact of cooking fumes. The following article can be useful: the impact of frying aerosol on human brain activity, https://doi.org/10.1016/j.neuro.2019.06.008.
Response: Thanks for your advice. We have read this article carefully and will discuss the potential nervous impact of cooking fumes from the perspective of this article. Our manuscript was revised based on your comments. The details were shown as Lines:
And a number of studies have proved that PM and VOCs emitted from cooking activities consisted of multiple hazardous chemical compounds which are easily deposited in the alveoli and can cause cardiovascular and respiratory disease or even to death. Apart from these negative effects, the cooking ultrafine particles can also impact on human brain activity [4-9]. Therefore, characterizing VOCs, PM and its chemical constituent and exploring beneficial method to reduce the emission of cooking fumes are significant in improving healthy environment of food processing or cooking for human beings.
- Reviewer’s comment: Page 2, paragraph 2,reference 11 never discussed the statements in the manuscript. Apparently, the authors cited to the wrong article, by mistake. Please correct it.
Response: Thanks for your carefully checking. We have carefully checked the manuscript and corrected the reference numbering.
- Reviewer’s comment: Page 2, paragraph 3, cites to Torkmahalleh et al. as reference 12, but this in the reference list, reference 12 is reserved for Peng et al.
Response: Thanks for your suggestions. Our manuscript was revised based on your comments.
- Reviewer’s comment: I believe there is a mistake in reference numbering in the reference list or in the text. Please fix it.
Response: Thanks for your carefully checking. We have carefully checked the manuscript and corrected the reference numbering.
- Peng, C.-Y.; Lan, C.-H.; Lin, P.-C.; Kuo, Y.-C. Effects of cooking method, cooking oil, and food type on aldehyde emissions in the cooking oil fumes. J. Hazard. Mater. 2017, 324, 160-167.
- Amouei Torkmahalleh, M.; Gorjinezhad, S.; Keles, M.; Unluevcek, H.S.; Azgin, C.; Cihan, E.; Tanis, B.; Soy, N.; Ozaslan, N.; Ozturk, F.; Hopke, P.K. A controlled study for the characterization of PM2.5 emitted during grilling ground beef meat. J. Aerosol. Sci. 2017, 103, 132-140.
- Reviewer’s comment: In section 2.3 no information about temperature.
Response: Thanks for your comment. We have revised the references of the manuscript according to your comment.
- Reviewer’s comment: According to Figure 1, please clarify that the samples taken after the quartz filter which are introduced to GC-MS and HPLC are free of particles or not. Other words are the data analyzed by HPLC and GC-MS are for only gas emissions or gas and particles?
Response: Thanks for your comment. This may be that we didn't express it clearly in the previous manuscript. the data of VOCs and carbonyl compounds we analyzed by HPLC and GC-MS are for only gas samples. Our experimental design aims to analyze volatile organic compounds and aldehyde ketone compounds in gas samples. The particles collected onto the quartz filters in cooking fumes were not used to analyze volatile organic compounds and aldehyde ketone compounds on particles. As for description of the method for collecting particles onto quartz filters, we failed to update content in time when writing this article. This is our mistake in revising the manuscript.
- Reviewer’s comment: Experiments in section 3.1, the authors discuss the order in mass concentration to be PM10 > PM4 > PM2.5 > PM1 and then cite it to reference 21. This order is quite obvious and expectable as in definition PM10 contains PM4, PM2.5, and PM1 and likewise PM4 contains PM2.5 and PM1. Thus, always this order has to be achieved. Otherwise, something is wrong with the instrument. I suggest to delete this discussion from the manuscript.
Response: Thanks for your recommendation. According to your recommendation, we have been modified it correctly. The modified content was shown as follows:
PM mass concentration emissions and size distribution for the group with additives and control group were indicated in Tab. 3. The statistically significant differences were observed between marinaded groups and control. It was found that particulate matter ranged from different size distribution emitted concentration with the highest level in the control. The results indicated that all marinate-treatment with white pepper powder, salt, garlic powder, MS can reduce PM concentration emissions compared to the control testing. The meat marinated with white pepper powder showed the largest reduction of PM mass concentration.
- Reviewer’s comment: Figure 3 does not really provide new information as figure 2 already covers the information in figure3. Please move it to the supplementary materials.
Response: Thanks for your recommendation. The related content was modified and moved the figure to the supplementary materials.
- Reviewer’s comment: Given the grilling period which was around 450 seconds, please explain what was the level of the grilling (medium or well-done).
Response: Thanks for your recommendation. The level of grilling was related to the time. According to the data obtained from our pre-experiment, the internal center temperature of meat can reach 85℃ when the time of grilling is about 420 seconds. Therefore, the time of grilling in our formal experiment was 420 seconds. At this grilling time, the meat products were already well-done.
- Reviewer’s comment:Apparently, the DustTrak was not calibrated against gravimetric mass measurements. DustTrak principle of working is light scattering which typically overestimates mass concentration. Thus, a side by side comparison between this device and gravimetric mass measurements using is needed to correct the concentrations. Author should create a section named as “ Limitation of this study” and explain the lack of this data in their manuscript.
Response: Thanks for your recommendation. We have supplemented a section about “Limitation of this study”. The supplementary contents are as follows:
2.7. Quality Control
The concentration of detected compounds in the DNPH-Silica cartridge blanks and stainless-steel sorbent tube were below the detection limits. The relative standard deviation of standard compounds was within 10 %. In addition, DustTrak Aerosol Monitor principle of working is light scattering which typically overestimates mass concentration and the readings of the DustTrak monitors were calibrated against a gravimetric sampler. In order to correct the error of instrument monitoring, gravimetric method is often used to measure the mass concentration of particulate matter. Due to the limitations of laboratory conditions, this was also the Limitation of this research.
- Reviewer’s comment:It objective of presenting figure 4 is not very well clear? What point the authors want to raise by this figure. I suggest they might be able to discuss it from exposure point. For example, people when they do grilling, they are present at the beginning of the grilling to adjust the position of the meat etc or two roll it over and then they may leave the grilling facility until it cooks itself. Thus, it is important to reduce the exposure at the early stages of the grilling when the chef is present over the stove. In this case, Figure 4 shows that marinating the meat will reduce the PM concentration significantly at the early stage of the grilling.
Response: Thanks for your recommendation. We have been discussed the results from exposure point. The modified content was shown as follows:
Fig. 2 presents that time profiles of PM emission during the meat-grilling process. As for the whole thermal process of grilling, it can be divided into three periods: initial stage, medium term, and final phase. The PM mass concentration showed a low emission level at the initial stage of grilling and it reached at maximum values from 350 to 390 sec in the medium term. At this stage, the mass concentration of particulate matter is much higher than the background concentration. During the medium stage of cooking activities,the average number concentrations of UFPs in the kitchen were about 20-40 times the background level [30]. At the final phase, the mass concentration of PM began to decline. Variation trend of particle mass concentration in the entire process of grilling shows that in the middle and late stage of cooking activities, human will be exposed to high concentration of PM environment.
- Reviewer’s comment:In Tables 4 and 5, it is not clear which pairs are statistically significant and which are not. The lower letter cases are confusing. Please discuss the statistically significant pairs, only between control and marinated ones (no need to discuss changes among the additives), in the main text and clarify which scenarios were statistically significant for the total carbonyls/VOCs and individual carbonyls/VOCs. Thus, the rest, which are not discussed, should be considered as insignificant. Then, from this point conclude if the additives had an impact on the total and individual carbonyl and VOCs or not.
Response: Thanks for your recommendation. Significant analysis of experimental data was performed by IBM SPSS Statistics 23 software and I do not understand what you mean about the confusion of lowercase letters. Values bearing different lowercase letters in the same line are significantly differences (p < 0.05). According to your recommendation, we have been discussed the statistically significant pairs. The modified content was shown as follows:
The effect of additives on the aldehyde and ketone compounds (13 representative compounds) emission was showed in table 5. Statistically significant differences were observed in the total carbonyl compounds between groups marinaded with salt, white pepper and control (P<0.05). For treatment groups marinaded with white pepper, salt, garlic powder, mixed spices, the average total carbonyl compounds concentrations were 7524.57±218.16 μg/m3, 6694.37±456.25 μg/m3, 7900.02±74.43 μg/m3, and 8206.65±269.55 μg/m3, respectively. The total aldehyde and ketone compounds were the highest emission for control group and reached as 8331.33±274.45 μg/m3 (Table 5).
As for the individual carbonyl compounds, formaldehyde, acetaldehyde, and acetone were the most abundant carbonyl compounds, with the percentages to total aldehyde and ketone compounds concentration of 24.40-34.01 %, 15.09-20.13 % and 16.85-27.86 %. But the acrolein were not detected in all the cooking fumes emitted from meat-grilling. Benzaldehyde, valeraldehyde, butyraldehyde and 2-butanone showed low percentage to total carbonyl compounds, and the percentages ranged from 0.46 % to 8.33 %. Also, other carbonyl compounds were determined in the cooking fumes, and their percentages to total carbonyl compounds were mostly < 10 %. Aldehyde compounds are the dominant odorous compounds generated from cooking activities [35]. The aldehyde compounds are produced via the hydrolyzation of the hydrocarbons in food and via the oxidation of fatty acids [36]. Fullana et al. found that volatile aldehydes were gengerated from β-scission of alkoxy radicals formed by the cleavage of fatty acid hydroperoxides [32-33].
- Reviewer’s comment:This study showed that additives particularly salt can reduce PM. Please discuss the reason beyond this observation. For example, PM should be the results of the nucleation of the low volatile organic compounds emitted during grilling for example high carbon carbonyl, etc. Can we conclude that additives reduced the emission of these types of compounds in the gas phase (as partly showed in Table 4 for salt ) and thus the nucleation rate of these compounds decreased during grilling and PM emission reduced? Basically the mechanisms for the impact of additives, particularly salts on the PM emissions are not discussed in this manuscript. The author might find the following article useful although it is related to the heating cooking oils.
Response: Thanks for your recommendation. We have tried our best to revise this content. We dare not draw conclusions about the mechanism of this content.
- Reviewer’s comment:Was any analyses conducted on the PM samples collected by the quartz filters? What was the purpose of collecting PM on the quartz filters? Only separating particles from gas phase before analyzing by GC-MS and HPLC? If yes, then please clearly state it in the manuscript. If this filter sampling was aimed for further analyses, then provide the data.
Response: Thanks for your recommendation. And the original purpose of sampling PM emitted from cooking fumes onto quartz fiber filters is to characterize the emission of polycyclic aromatic hydrocarbons on PM and observe the microscopic morphology of particulates. The morphologies and compositions of particles adsorbed onto quartz fiber filters were investigated by field emission scanning electron microscope (FESEM) combined with energy dispersive X-ray (EDX) instruments. However, we consider that this part of data is not suitable for the theme of this manuscript, so we did not edit this part of data in the manuscript.
- Reviewer’s comment:Following the previous comments, the authors showed that the emissions were analyzed before and after PM collection on the quartz filters. This means the GC and HPLC results should be presented before and after PM collections. Other words, the authors should show what were the VOCs and carbonyl contents of the emissions for (gas + PM) and (only gas). Then they can conclude the VOCs and carbonyl content of only PM. They can then discuss the impact of additives on the PM carbonyl and VOCs (total and individual) as well and compare the results with the control and discuss the statistically significant cases. However, this requires the assumption that quartz filter remove the totally the PM before the emissions being introduced to the GC and HPLC.
Response: Thanks for your recommendation. Sorry, we did not analyze before PM collection on the quartz filters by GC and HPLC. The quartz filter membrane without collecting particles is a blank group, and there is no component on the filter membrane to be analyzed. This may be that we didn't express it clearly. And the emission of volatile organic compounds and carbonyl contents only for gas samples. Our experimental design aims to analyze volatile organic compounds and aldehyde ketone compounds in gas samples. The particles collected onto the quartz filters in cooking fumes were not used to analyze volatile organic compounds and aldehyde ketone compounds on particles.
The purpose of sampling particles onto quartz fiber filters described in our previous manuscript is to analyze the emission characteristics of polycyclic aromatic hydrocarbons adsorbed on particles in cooking fumes. Also, the micro morphology and elemental composition of particles collected onto the quartz filters were analyzed by cold field emission scanning electron microscope and energy dispersive spectrometer.
- Reviewer’s comment:The authors did not discuss why additives (except salt) showed higher VOCs compared to the control case although only one case showed statistically significant difference. Canwe conclude additives (except salt) examined in this study themselves contribute to the VOCs and Carbonyl? As a results when additives were used, VOCs increased? Please discuss and cite properly, the VOCs and possibly carbonyl emissions from the heated additives.
Response: Thanks for your recommendation. We have revised the article from your point of view: “the VOCs and possibly carbonyl emissions from the heated additives”.
The modified content was shown as follows:
As for the individual carbonyl compounds, formaldehyde, acetaldehyde, acetone, and Propionaldehyde were the most abundant carbonyl compounds, with the percentages to total aldehyde and ketone compounds concentration of 24.40-34.01 %, 15.09-20.13 %, 16.85-27.86 %, and 6.00-9.60 %. And pork contains various unsaturated fatty acids and saturated fatty acids [32-33], linolenic acid was associated with production of acetaldehyde, oleic acid was associated with production of acetaldehyde and propanal, and palmitic acid was associated with production of nonanal [12].
Although the total VOCs of grilling meat premarinated with salt showed an ability in reducing emission as compared with the control group (476.67±110 μg/m3), there was no statistically significant difference observed (p > 0.05). Among the other three pre-curing treatments, only the marinaded group significantly increased the emissions concentration of total volatile organic compounds (p < 0.05). Spice additives (except salt) contain some natural antioxidant substances [37]. These substances will also undergo complex chemical reaction like fatty acids and protein in meat due to high temperature in the grilling [1]. Generally, saturated fatty acids are more resistant to oxidation than unsaturated ones. Although the content of saturated fatty acids in pork is higher than unsaturated fatty acids, they also can be oxidized to produced cleavage products when the temperature of thermal processing exceed 150 ℃ [38].
- Reviewer’s comment:Please discuss the results of your study with the following reference that showed addition of salt to the meat may increase the PM emissions.
(O'Leary, C., de Kluizenaar, Y., Jacobs, P., Borsboom, W., Hall, I., & Jones, B. (2019). Investigating measurements of fine particle (PM 2.5) emissions from the cooking of meals and mitigating exposure using a cooker hood. Indoor air, 29(3), 423-438.)
Response: Thanks for your recommendation. Thanks for your questions and advice. We have read this article carefully and will discuss the results of PM from the perspective of this article. Our manuscript was revised based on your comments.
- Reviewer’s comment: The last name of the author “Torkmahalleh” in the text and reference list has to be changed to the correct last name “Amouei Torkmahalleh”. Thus, it should be cited as Amouei Torkmahalleh et al. and listed properly in the reference list.
Response: Thanks for your carefully checking. We have carefully checked the last name of authors in the manuscript and corrected it.
Reviewer 2 Report
Although in many comments the authors say that the manuscript was "revised based on comments" I have noticed that some of them were not addressed properly. For example, in the comment #7, I suggest to include the description of how the VOCs were identified. They say that standards were used for quantification, but all of these are not frlly described in the text. The authors did not answer when I have asked if all the compounds really were in all treatments.
In figure 2, it is not clear which the letters are about in the statistics in the figure. The graphs in figure 4 are too small and "A" is in the bottom of the figure.
I think in the case of state that moisture and lipid content of each batch is the same, and if the authors say that these parameteres were not measured, so it can´t be said in the text. I Suggest then exclude this statement.
A revision of English language is necessary yet, as in some places it is not written in good English. Beyond writing style is odd in some places, the word "compound marinade" seems odd too. Still de use of "sousing", although the authors have explained what it was, but I is the usage correct?
Author Response
Response to reviewer 2
Thank you so much for your comments and suggestions. Our manuscript has been revised based on your comments. The details are shown as following.
Responds to the reviewer’s comments:
Reviewer #2:
- 1. Reviewer’s comment:For example, in the comment #7, I suggest to include the description of how the VOCs were identified. They say that standards were used for quantification, but all of these are not fully described in the text.
Response: Thanks for your comment. The methodology of qualitative analysis and quantification of 22 representative VOCs was “the retention time and mass spectrum were compared for qualitative analysis and quantification of 22 representative VOCs based on the standard curve”. our manuscript was revised based on your comments. The details were shown as follows:
The individual concentrations of 22 representative VOCs were quantified using the standard compounds, the method was as follows: firstly, remove 25 μL, 50 μL, 100 μL, 250 μL and 500 μL of the standard stock solution with microsyringe to 10 mL volumetric flask and dilute it with methanol to the mark line, The concentration gradients of 5.00, 10.0, 20.0, 50.0 and 100 μg / mL mixed standard solution were prepared. Installing the adsorption tube on the thermal desorption standard sample loading platform, and injecting 1.0 μL the mixed standard solution with microsyringe into the blank adsorption tube. Then adding internal standard solution into the adsorption tube at the same time, purging the adsorption tube with N2 for 5 min, Removing the adsorption tube and sealing both ends of the adsorption tube with sealing cap to obtain a calibration series of adsorption tubes with contents of 5.00, 10.0, 20.0, 50.0 and 100 ng. Secondly, Putting the adsorption tubes of the calibration curve series into the thermal desorption instrument, analyzing from low concentration to high concentration according to the test conditions. Then drawing the calibration curve with the least square method or relative response factor. The retention time and mass spectrum were compared for qualitative analysis and quantification of 22 representative VOCs based on the standard curve.
- Reviewer’s comment:The authors did not answer when I have asked if all the compounds really were in all treatments.
Response: Thanks for your comment. The cooking fumes produced from the grilling were sampled by stainless steel sampling tube filled with adsorbent (involving Carbopack C sorbent, Carbopack B sorbent, and Carboxen 1000 sorbent). Then the samples were determined using a thermal desorption system-gas chromatography coupled to a mass spectrometry. A total of 22 target compounds were selected. Through the retention time and mass spectrum were compared for qualitative analysis, we determined the target VOCs apart from Isopropyl alcohol and Hexamethyldisiloxane.
Of course, 22 target compounds were part of the chemical compounds in the cooking fumes. According to total ion flow chromatographic separation figure, a total of 53 peak, identified 40 compounds. The 22 target compounds were selected according to the national standard.
- Reviewer’s comment:In figure 2, it is not clear which the letters are about in the statistics in the figure. The graphs in figure 4 are too small and "A" is in the bottom of the figure
Response: Thanks for your recommendation. In order to clearly express the significance of the data in the figure 2, we have replaced the figure2 with table 3. The details were shown as follows:
Table 3. Mass concentration and size distribution of PM1.0, PM2.5, PM4.0, PM10 emitted from the cooking fumes
|
Additive types |
Mass concentration |
|||
|
PM1.0 |
PM2.5 |
PM4.0 |
PM10 |
|
|
Control |
36.20±5.28a |
37.06±5.45a |
37.73±5.54a |
39.03±5.95a |
|
Salt |
18.33±3.18bc |
18.63±3.20bc |
19.00±4.12bc |
20.06±2.93bc |
|
White pepper |
12.50±4.48d |
12.78±4.13d |
12.96±5.16d |
13.73±4.14d |
|
Garlic powder |
23.13±2.65b |
23.50±2.30b |
24.03±3.02b |
25.87±3.32b |
|
MS |
15.20±0.43cd |
15.47±0.97cd |
15.73±0.49cd |
16.87±1.36cd |
Mass concentration unit: mg/m3.Values bearing different lowercase letters in the same column are significantly differences (p < 0.05). Error bar denotes standard deviation. each bar means significant differences. Error bar denotes standard deviation.
The size and format of graphs meet the requirements of the journal, The reason why the graphs in figure 4 are too small mainly was limited by typesetting requirements. The order of the four graphs in Figure 4 has been adjusted according to your suggestions. The details were shown in the manuscript.
- Reviewer’s comment:I think in the case of state that moisture and lipid content of each batch is the same, and if the authors say that these parameters were not measured, so it can´t be said in the text. I Suggest then exclude this statement.
Response: Thanks for your recommendation. We have supplemented the test data and methods of moisture content, lipid content and protein content of meat. The supplementary contents are as follows:
2.3. Determination the chemical composition of the pork
As for moisture content, lipid content and protein content of raw pork and marinated meat, each batch was determined by the following method: (1) moisture content: the determination of moisture content in pork mainly referred to the direct drying method in the GB/T 5009.3-2016[21], the experimental procedures are as follows: firstly, leave the cover off the weighing bottle and put them into drying oven until dry them to constant weight. The constant weight of weighing bottle was recorded to M0. Then the 5.00 g minced meat of pork belly was weighed and put into weighing bottle, the total weight of meat and weighing bottle was recorded to M1, Secondly, place the sample and weighing bottle in 105 ℃ drying oven until dry them to constant weight. After being taken out, it should be placed in to a dryer to cool for 30 min. The constant weight of the sample and weighing bottle was recorded to M2. The moisture content should be calculated according to the formula:
Moisture content(%) = M2-M0/M1-M0×100%
(2) lipid content: the determination of lipid content in pork mainly referred to GB/T 5009.6-2016[22]-soxhlet extractor method. (3) protein content: the determination of lipid content in pork mainly referred to GB/T 5009.5-2016-Kjeldahl nitrogen determination method [23], and adjust the determination of protein content in combination with the instrument instructions of Kjeldahl nitrogen determination instrument.
The experimental data moisture content, lipid content and protein content were shown in Table S1 in the supplementary data.
- Reviewer’s comment:A revision of English language is necessary yet, as in some places it is not written in good English. Beyond writing style is odd in some places, the word "compound marinade" seems odd too. Still de use of "sousing", although the authors have explained what it was, but I is the usage correct?
Response: Thanks for your recommendation. The term "compound marinade" was has been modified to mixed spices (MS). The mixed spices (MS) were complex, and through mixing the salt, white pepper and garlic powder by a ratio of 1:2:2, the mixed spices (MS) were made to marinade the meat. We have been used it throughout the text. The modified content was shown as follows:
In addition, through mixing the salt, white pepper and garlic powder by a ratio of 1:2:2, the mixed spices (MS) were made to marinade the meat.
Reviewer 3 Report
The authors have not addressed most of the points made by the reviewer. Therefore, this work still needs to be reviewed in depth.
General comments
- Put the manuscript in the journal format. On the journal's website (https://www.mdpi.com/journal/foods/instructions) you can download a template in which the line numbers appear.
- The authors did not respond to the following comment: “I miss the determination of the chemical composition of each of the batches studied, since it would allow to justify some of the results found”.
Specific comments
Abstract
- The term "compound marinade" is not used correctly since it seems to make no sense. Please rewrite it correctly or use a term that better defines this batch. Use it throughout the text.
- Define "TPM in the abstract.
Introduction
- The authors did not understand the comment “Reflect in the introduction that chemical reactions occurred during cooking could change meat final quality and consumer acceptability” since they mentioned that “The addition of various additives can also improve the quality of meat products and consumer acceptability”. Please, explain how chemical reactions occurred during cooking could change meat final quality and consumer acceptability.
- The citation of the reference “Amouei Torkmahalleh et al” is not correct. Please, replace it by “Torkmahalleh et al. [13]”.
Material and methods
- Section 2.2. should begin by showing which batches were used in the study, including the information showed below about the control batch (“As a control, the same lean pork was conducted without any spice additive”).
Results and Discussion
- The results have not been discussed using more recent references.

Author Response
Response to reviewer 3
Thank you so much for your comments and suggestions. Our manuscript has been revised based on your comments. The details are shown as following.
Responds to the reviewer’s comments:
Reviewer #3:
- 1. Reviewer’s comment: I miss the determination of the chemical composition of each of the batches studied, since it would allow to justify some of the results found.
Response: Thanks for your comment. I did not fully understand what indicators you mean by the chemical composition of each of the bathes. But we have supplemented the experimental data of fat content, moisture content and protein content of raw pork. Also, the data of fat content, moisture content and protein content of raw pork marinaded with different additives have been supplemented in our manuscript. The details were shown as follows:
As for moisture content, lipid content and protein content of raw pork and marinated meat, each batch was determined by the following method: (1) moisture content: the determination of moisture content in pork mainly referred to the direct drying method in the GB/T 5009.3-2016[21], the experimental procedures are as follows: firstly, leave the cover off the weighing bottle and put them into drying oven until dry them to constant weight. The constant weight of weighing bottle was recorded to M0. Then the 5.00 g minced meat of pork belly was weighed and put into weighing bottle, the total weight of meat and weighing bottle was recorded to M1, Secondly, place the sample and weighing bottle in 105 ℃ drying oven until dry them to constant weight. After being taken out, it should be placed in to a dryer to cool for 30 min. The constant weight of the sample and weighing bottle was recorded to M2. The moisture content should be calculated according to the formula:
Moisture content (%) = M2-M0/M1-M0×100%
(2) lipid content: the determination of lipid content in pork mainly referred to GB/T 5009.6-2016[22]-soxhlet extractor method. (3) protein content: the determination of lipid content in pork mainly referred to GB/T 5009.5-2016-Kjeldahl nitrogen determination method [23], and adjust the determination of protein content in combination with the instrument instructions of Kjeldahl nitrogen determination instrument.
The experimental data moisture content, lipid content and protein content were shown in Table S1 in the supplementary data.
Table S1 Comparison of chemical composition of pork belly marinaded with various additives
|
Chemical composition |
Raw pork |
Additive types |
|||
|
Salt |
White pepper |
Garlic powder |
MS |
||
|
Moisture content |
71.25±1.31a |
64.57±1.23b |
61.51±0.34c |
68.47±1.82a |
65.74±1.01b |
|
Fat content |
6.84±1.93a |
5.87±0.85b |
6.56±1.01a |
6.83±1.29a |
6.64±1.83a |
|
Protein content |
20.21±0.99a |
19.55±1.53a |
19.83±1.17a |
19.01±1.03a |
19.36±0.87a |
- Reviewer’s comment:The term "compound marinade" is not used correctly since it seems to make no sense. Please rewrite it correctly or use a term that better defines this batch. Use it throughout the text.
Response: Thanks for your recommendation. The term "compound marinade" was has been modified to mixed spices (MS). The mixed spices (MS) were complex, and through mixing the salt, white pepper and garlic powder by a ratio of 1:2:2, the mixed spices (MS) were made to marinade the meat. We have been used it throughout the text. The details were shown as follows:
2.1. Materials, and sampling instruments
Four additives including salt, white pepper and garlic powder are commonly used in the grilling of meat and were purchased from local supermarket in HeFei, P.R, China. Pork belly and high density bamboo charcoal (HDBC) also purchased from a local Carrefour supermarket (HeFei, P.R, China). In addition, through mixing the salt, white pepper and garlic powder by a ratio of 1:2:2, the mixed spices (MS) were made to marinade the meat.
- Reviewer’s comment:Define "TPM in the abstract.
Response: Thanks for your recommendation. The meaning of “TPM” is total particles mass concentration, we abbreviated “total particles mass concentration” to “TPM”. We have revised the abstract and defined “TPM” in the abstract in the manuscript according to your recommendation. The details were shown as follows:
Results showed that application of white pepper, salt, garlic powder, and mixed spices (MS)could significantly reduce the total particles mass concentration (TPM) emissions during meat-grilling by 65.07 %, 47.86 %, 32.87 %, 56.01 %, respectively.
- Reviewer’s comment:The authors did not understand the comment “Reflect in the introduction that chemical reactions occurred during cooking could change meat final quality and consumer acceptability” since they mentioned that “The addition of various additives can also improve the quality of meat products and consumer acceptability”. Please, explain how chemical reactions occurred during cooking could change meat final quality and consumer acceptability.
Response: Thanks for your comment. The manuscript was revised based on your comments. The details were shown as follows:
Additives are widely applied in the different cooking methods. Synergistic effect of spices additive and different cooking methods can cause the formation of characteristic aroma via induced reactions that ameliorate the aroma profiles of products. The key aroma compounds will be generated in the thermal process of cooking activities due to a wide range of complex chemical reactions involving the lipid oxidation and pyrolysis reactions, thiamine degradation, proteolyisis reactions, Maillard reaction, and Maillard-lipid interactions. [16-19] The addition of various additives can improve the formation of key aroma compounds to change the quality of meat products and consumer acceptability.
- Reviewer’s comment:The citation of the reference “Amouei Torkmahalleh et al” is not correct. Please, replace it by “Torkmahalleh et al. [13]”
Response: Thanks for your recommendation. We didn't make a careful distinction between last names and first names. According to your recommendation, we have been modified it correctly. The details were shown as follows:
Torkmahalleh et al. [13] studied the effect of additives on the emission characterization of PM2.5 and total particle number during heating cooking oils. The results indicated that addition of sea salt could reduce the PM2.5 concentration by 86 - 91% and total particle number by 45 - 53% as compared with control group. Katragadda et al. [14] assessed the impacts of oil types on volatile aldehydes emissions produced from heated cooking oils.
- Reviewer’s comment:Section 2.2. should begin by showing which batches were used in the study, including the information showed below about the control batch (“As a control, the same lean pork was conducted without any spice additive”).
Response: Thanks for your comment. Our manuscript was revised based on your comments. The details were shown as follows:
2.2. PM and VOCs emission experiments for different additives
Four additives were selected to marinade: white pepper powder, garlic powder, salt, and mixed spices. The pork was pre-marinated with additives at a ratio of 0.5g/100g, respectively. Then the marinated meat was transferred to low temperature cold store, and marinaded for 6 hours. As a control, the same pork was conducted without any spice additive. The marinated pork was grilled using a round grilling plate. The HDBC was used as the fuel of meat-grilling process, and transferred to the charcoal basin after fully burned for 5min. Besides, a thermocouple was placed above the grilling plate to monitor the heating temperature, and the variations of temperature were less than 6 ℃. When the temperature of grilling plate reached 320 ℃, the simulation meat-grilling started.
In order to determine the emission characteristics of PM and VOCs objectively and scientifically during the meat-grilling in the presence of different additives, our group have established the simulation and detection platform for cooking fumes emitted from meat-grilling. The meat-grilling experiments were conducted at HeFei University of Technology, FeiCui Lake campus. The cooking fumes emitted from the grilling were captured by a 40-inch by 50-inch stainless steel hood and ducted to the stack of the facility with an exhaust fan. The exhaust fan has a variable speed drive and controller, which was used to adjust the velocity and flow rates through the stack. The experiments system used to conduct this study is shown in Fig. 1.
- Reviewer’s comment:The results have not been discussed using more recent references.
Response: Thanks for your comment. We have made a lot of revises to the results and discussion of the article. The modified content was shown as “Results and Discussion”.
Reference:
- State Health and Family Planning Commission of the people's Republic of China. National food safety standard-Determination of moisture in foods. 2016.
- State Health and Family Planning Commission of the people's Republic of China. National food safety standard-Determination of fat in foods. 2016.
- State Health and Family Planning Commission of the people's Republic of China. National food safety standard-Determination of protein in foods. 2016.